# The causal effect of obesity on prediabetes and insulin resistance reveals the important role of adipose tissue in insulin resistance

Zong Miao[1,2], Marcus Alvarez[1], Arthur Ko[3], Yash Bhagat[1], Elior Rahmani[4], Brandon Jew[2,4], Sini Heinonen[5], Linda Liliana Muñoz-Hernandez[6,7,8], Miguel Herrera-Hernandez[9], Carlos Aguilar-Salinas[6,7,8], Teresa Tusie-Luna[10], Karen L. Mohlke[11], Markku Laakso[12], Kirsi H. Pietiläinen[5,13], Eran Halperin[1,4,14,15,16], Päivi Pajukanta[1,2,16]*

1 Department of Human Genetics, David Geffen School of Medicine at UCLA, Los Angeles, California, United States of America, 2 Bioinformatics Interdepartmental Program, UCLA, Los Angeles, California, United States of America, 3 Department of Medicine, David Geffen School of Medicine at UCLA, Los Angeles, California, United States of America, 4 Computer Science Department in the School of Engineering, UCLA, Los Angeles, California, United States of America, 5 Obesity Research Unit, Research Program for Clinical and Molecular Metabolism, Faculty of Medicine, University of Helsinki, Helsinki, Finland, 6 Unidad de Investigación en Enfermedades Metabólicas, Dirección de Nutrición, Instituto Nacional de Ciencias Médicas y Nutrición Salvador Zubirán, Mexico City, Mexico, 7 Departamento de Endocrinología y Metabolismo, Instituto Nacional de Ciencias Médicas y Nutrición Salvador Zubirán, Mexico City, Mexico, 8 Tecnologico de Monterrey, Escuela de Medicina y Ciencias de la Salud, Monterrey, Nuevo Leon, México, 9 Departamento de Cirugía, Instituto Nacional de Ciencias Médicas y Nutrición, Mexico City, Mexico, 10 Unidad de Biología Molecular y Medicina Genómica Instituto de Investigaciones Biomédicas UNAM / Instituto Nacional de Ciencias Médicas y Nutrición Salvador Zubiran, Mexico City, Mexico, 11 Department of Genetics, University of North Carolina, Chapel Hill, North Carolina, United States of America, 12 Institute of Clinical Medicine, Internal Medicine, University of Eastern Finland and Kuopio University Hospital, Kuopio, Finland, 13 Obesity Center, Endocrinology, Abdominal Center, Helsinki University Central Hospital and University of Helsinki, Helsinki, Finland, 14 Department of Computational Medicine, UCLA, Los Angeles, California, United States of America, 15 Department of Anesthesiology and Perioperative Medicine, UCLA, Los Angeles, California, United States of America, 16 Institute for Precision Health, David Geffen School of Medicine at UCLA, Los Angeles, California, United States of America

* ppajukanta@mednet.ucla.edu

**Data Availability Statement:** This research has been conducted using the UK Biobank Resource under Application Number 33934. The UK Biobank data is available from the UK Biobank data

## Abstract

Reverse causality has made it difficult to establish the causal directions between obesity and prediabetes and obesity and insulin resistance. To disentangle whether obesity causally drives prediabetes and insulin resistance already in non-diabetic individuals, we utilized the UK Biobank and METSIM cohort to perform a Mendelian randomization (MR) analyses in the non-diabetic individuals. Our results suggest that both prediabetes and systemic insulin resistance are caused by obesity ($p = 1.2×10^{-3}$ and $p = 3.1×10^{-24}$). As obesity reflects the amount of body fat, we next studied how adipose tissue affects insulin resistance. We performed both bulk RNA-sequencing and single nucleus RNA sequencing on frozen human subcutaneous adipose biopsies to assess adipose cell-type heterogeneity and mitochondrial (MT) gene expression in insulin resistance. We discovered that the adipose MT gene expression and body fat percent are both independently associated with insulin resistance ($p \leq 0.05$ for each) when adjusting for the decomposed adipose cell-type proportions. Next, we showed that these 3 factors, adipose MT gene expression, body fat percent, and adipose

repository, but restrictions apply to the availability of these data, which were used under license for the current study and therefore are not publicly available. Data are however available with the permission of UK Biobank. The GTEx dataset analyzed during this study are available in the dbGAP repository, phs000424.v6.p1 (currently available under accession number phs000424.v8. p2). The METSIM gene expression data were previously made available in GEO under GSE135134.

**Funding:** This study was funded by National Institutes of Health (NIH) grants HL-095056, HL-28481, R01HG010505, and U01 DK105561. Z.M was supported by the AHA grant 19PRE34430112, M.A. was supported by the HHMI Gilliam grant, and A.K. by the NIH grant F31HL127921. E.H., E.R. and B.J. were partially supported by the National Science Foundation grant 1705197. E.H., E.R., and B.J. were partially supported by NIH/NHGRI HG010505-02. K.H.P. was supported by the Academy of Finland (272376, 266286, 314383, 315035), Finnish Medical Foundation, Finnish Diabetes Research Foundation, Novo Nordisk Foundation, Gyllenberg Foundation, Sigrid Juselius Foundation, Helsinki University Hospital Research Funds, Government Research Funds and University of Helsinki. The funders had no role in study design, data collection, and analysis, decision to publish, or preparation of the article.

**Competing interests:** The authors have declared that no competing interests exist.

cell types, explain a substantial amount (44.39%) of variance in insulin resistance and can be used to predict it ($p \leq 2.64 \times 10^{-5}$ in 3 independent human cohorts). In summary, we demonstrated that obesity is a strong determinant of both prediabetes and insulin resistance, and discovered that individuals' adipose cell-type composition, adipose MT gene expression, and body fat percent predict their insulin resistance, emphasizing the critical role of adipose tissue in systemic insulin resistance.

## Author summary

Obesity is a global health epidemic predisposing to type 2 diabetes (T2D) and other cardiometabolic disorders. Previous studies have shown that obesity has a causal effect on T2D; however, it remains unknown whether obesity causes prediabetes and insulin resistance already in non-diabetic individuals. By utilizing almost half a million individuals from the UK Biobank and the Finnish METSIM cohort, we identified a significant causal effect of obesity on prediabetes and insulin resistance among the non-diabetic individuals. Next, we investigated the role of subcutaneous adipose tissue in these obesogenic effects. We discovered that the adipose mitochondrial gene expression and body fat percent are independently associated with insulin resistance after adjusting for the tissue heterogeneity. For the latter, we estimated the adipose cell type proportions by utilizing single-nucleus RNA sequencing of frozen adipose tissue biopsies. Moreover, we established a prediction model to estimate insulin resistance using body fat percent and adipose RNA-sequencing data, which enlightens the importance of adipose tissue in insulin resistance and provides a helpful tool to impute the insulin resistance for existing adipose RNA-sequencing cohorts. Overall, we discover the potential causal effect of obesity on prediabetes and insulin resistance and the key role of adipose tissue in insulin resistance.

## Introduction

The global obesity epidemic is driving the concomitant rapid increase in the prevalence of cardiometabolic disorders, including type 2 diabetes (T2D) [1,2]. It is well established that obesity, prediabetes, and insulin resistance are tightly associated [3–8]. Although prediabetes is an inevitable stage for T2D patients, some people with prediabetes do not ultimately develop diabetes given that the annualized conversion rate of prediabetes to diabetes is estimated to be 5%–10% [9]. Thus, it is important to investigate whether obesity is also a causal risk factor for the early stages of T2D development (i.e. prediabetes). Moreover, inflammation has been identified as the link between obesity and insulin resistance [10–15]. For example, Roberts-Toler et al. showed that diet-induced obesity can cause insulin resistance in mouse brown adipose tissue [16]. Wensveen et al. showed that natural killer cells can mediate the association between obesity and insulin resistance in mice [17]. However, the direction of the causal effect between obesity and insulin resistance remains elusive in humans [18,19]. Thus, direct evidence of obesity causing systemic insulin resistance in humans is still lacking. To this end, we performed a Mendelian randomization (MR) analysis using the genotype and metabolic traits of unrelated non-diabetic individuals from both UK Biobank (UKB) [20] and the Finnish METabolic Syndrome In Men (METSIM) cohort [21]. This report suggests for the first time using MR that obesity (i.e. body fat percent) is causally related with both prediabetes and insulin resistance in non-diabetic humans.

The key functions of adipose tissue, i.e. lipogenesis (storing fat) and lipolysis (mobilizing the stored fat), make it one of the most important tissues contributing to obesity. Thus, it would be important to better understand how much this endocrine tissue contributes to insulin resistance and T2D. Since adipose tissue is complex and contains multiple cell-types, adipose cell-type composition may be affected by obesity. Weisberg et al. showed that the number of macrophages increases in the adipose tissue of obese mice [22]. Furthermore, in human adipose tissue, high BMI was reported to be negatively correlated with the number of adipocytes [23] and positively correlated with the size of the adipocytes [22–24]. However, the effects of different adipose cell-type proportions on insulin resistance have not been systematically assessed in humans previously.

Fluorescence-activated cell sorting (FACS) has been used for characterizing and defining some of the cell types in human adipose samples [25–27]. Even though the number of marker proteins that can be simultaneously measured in FACS has progressively increased [28], FACS relies on predetermined cell-type specific marker proteins to isolate different cell types and is thus unable to discover new cell types or sub-cell types. Moreover, since FACS requires a large starting number of cells (more than 10,000) in suspension [29], it is unable to isolate single cells from a low quantity cell population. Overall, it is highly challenging to evaluate all cell types of solid tissues, such as adipose tissue, using FACS. Thus, to thoroughly investigate the tissue heterogeneity in human adipose tissue, we performed single nucleus RNA sequencing (sn-RNA-seq) of all adipose cell types using frozen human subcutaneous adipose biopsies. We then utilized the sn-RNA-seq data to define expression profiles of signature genes in different adipose cell-types to decompose cell-type proportions in the bulk adipose RNA-seq cohorts. This helped us leverage the gene expression information available in the adipose bulk RNA-seq data to assess whether adipose cell-type composition influences systemic insulin resistance.

Previous studies have shown that the biogenesis and metabolic activities of the mitochondria (MT) are impaired in the adipose tissue of obese individuals [2, 30–33]. Experimental evidence also shows that declined MT function can elicit insulin resistance in mice [34,35]. Paglialunga et al. further demonstrated that elevated MT reactive oxygen species (ROS) emission in murine white adipose tissue contributes to insulin resistance [35]. Furthermore, dysfunction of MT in muscle and liver associates with insulin resistance in humans [36,37]. However, since the MT activity, obesity status (body fat percent or BMI), and systemic insulin resistance are associated with each other, it is unclear whether these associations are caused by independent mechanisms or confounded by a shared trait. To this end, we investigated whether MT gene expression in human adipose tissue is independently associated with systemic insulin resistance and body fat percent/BMI. Furthermore, we built a prediction model to investigate whether systemic insulin resistance (i.e. assessed using the Matsuda index) in humans can be predicted using adipose cell-type proportions, adipose MT gene expression, and body fat percent as an input. Overall our studies helped determine the causal role of obesity in human insulin resistance, of which a major portion is driven by adipose tissue.

## Results

### Evidence from Mendelian randomization for the role of obesity in prediabetes and systemic insulin resistance in non-diabetic individuals

Although obesity and prediabetes are known to be associated [7,8], there is no previous MR evidence about the causal direction between them in non-diabetic individuals. To this end, we performed an MR analysis to investigate whether prediabetes (assessed by serum HbA1C level between 5.7–6.4 [38]) is caused by obesity. Fig 1A shows the two-sample MR models we used to explore causal associations between prediabetes and obesity (body fat percent).

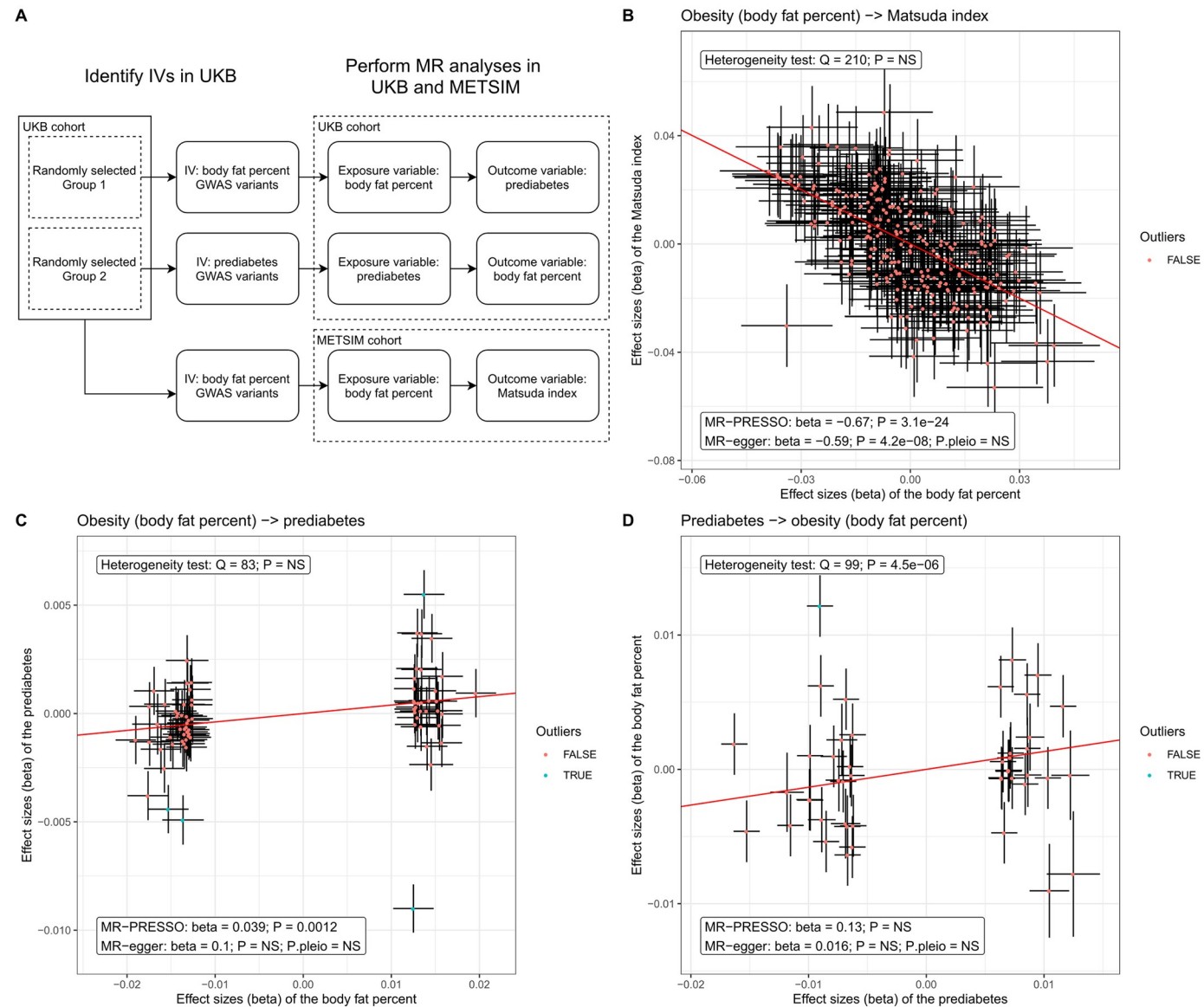

**Fig 1. MR analysis shows the causal relationship of body fat percent on prediabetes and Matsuda index, i.e. obesity leads to insulin resistance.** (A) Workflow of our MR analysis in UKB and METSIM cohorts. (B) The variant effect sizes on the exposure (i.e. body fat percent) are associated with the variant effect sizes on the outcome (i.e. the Matsuda index). The slope indicates the estimated causal effect of the exposure on the outcome. The label boxes showed the MR results after MR-PRESSO adjusted for potential pleiotropy. P.pleio indicates the p-value of pleiotropy identified by MR-egger. The IVs are significant body fat percent GWAS variants in UKB (see the Methods for details). (C) A two-sample MR analyses show the causal effect of obesity (i.e. body fat percent) on prediabetes in UKB cohort. The IVs are significant prediabetes GWAS variants in UKB (see the Methods for details). (D) A two-sample MR analysis did not identify the causal effect of prediabetes on obesity (i.e. body fat percent). The p-values of the causal effects from MR-PRESSO and MR-egger were adjusted for 3 tests using a Bonferroni correction. To keep a stringent control of the potential pleiotropy, the other p-values (Heterogeneity test and P.pleio) were not adjusted for multiple testing. NS indicates a non-significant p-value (p-value > 0.05). The IVs are significant body fat percent GWAS variants in UKB (see the Methods for details).

As BMI reflects both fat and lean mass, we used body fat percent instead of BMI in our MR analyses to utilize a more specific surrogate of obesity [39,40]. For this MR analysis, we first randomly separated the UKB cohort into two independent groups (n = ~190k in each group). Then we performed a GWAS for body fat percent in one of the UKB groups that identified 74 non-redundant SNPs ($R^2 < 0.01$) significantly associated with body fat percent

($p < 5 \times 10^{-8}$), which we used as the genetic instrumental variables (IVs) in the MR analysis (see Methods). Next, we used the second UKB group to perform a GWAS analysis on prediabetes to fulfill the two-sample MR design. Since dyslipidemia and abnormal blood pressure are known factors associated with obesity and diabetes [41,42], we excluded the IVs that are significantly associated with diastolic/systolic blood pressure or serum total triglycerides in the UKB cohort to avoid potential pleiotropy. Finally, we used the summary statistics from the GWAS analysis and MR-PRESSO [43] to identify a significant positive causal effect of body fat percent on prediabetes in the UKB (estimate effect = 0.039; adjusted p-value = $1.2 \times 10^{-3}$). To further investigate the effect of obesity on prediabetes, we also employed MR-egger and heterogeneity test on the IVs adjusted by MR-PRESSO. Although MR-egger did not identify a significant causal effect of body fat percent on prediabetes using the threshold of p-value < 0.05, neither MR-egger nor heterogeneity test identified a significant pleiotropy in this MR analyses (Fig 1C). Since MR-egger usually tend to have less power compared to other MR methods [44], finding no evidence for pleiotropy from MR-egger and heterogeneity test appears still to be in line with the causal effect of body fat percent on prediabetes identified by MR-PRESSO (Fig 1C). This also suggests that our MR model fulfills the second and third assumptions of MR analysis (i.e. the IV is not associated with the hidden confounders or with the outcome variable (i.e. prediabetes) when conditioning on the exposure variable (i.e. body fat percent)). To test the possibility of the reverse causal path, we utilized 47 independent prediabetes GWAS SNPs as IVs from our GWAS in the second half of the UKB and explored the potential causal effect of prediabetes on obesity. Fig 1D shows that there is no causal effect of prediabetes on body fat percent (estimated effect = 0.13, adjusted p-value > 0.05). As we performed 3 MR analyses to explore the causal effects among obesity, insulin resistance and prediabetes, we adjusted the p-values of causal effects for 3 tests using a Bonferroni correction. Thus, using MR in the UKB, we established a one-way causal effect of obesity on prediabetes.

To further investigate this finding, we next explored the causal relationship between obesity and systemic insulin resistance (i.e. decreased insulin sensitivity assessed by the Matsuda index) in the METSIM cohort. We used the 241 significant non-redundant body fat percent GWAS SNPs identified in the UKB as IVs for the MR analyses in METSIM. Using MR-PRESSO, we discovered a negative causal effect of body fat percent on the Matsuda index (i.e. insulin sensitivity) in METSIM (estimate effect = -0.67, p-value = $3.1 \times 10^{-24}$). It is worth noting that MR-PRESSO did not find any evidence of global pleiotropy (p-value > 0.05) when testing the causal effect of body fat percent on the Matsuda index in METSIM (Fig 1B). Moreover, MR-egger verified this finding (adjusted p-value = $4.2 \times 10^{-8}$) while it identified no evidence of pleiotropy (P.pleio > 0.05). A heterogeneity test also showed the concordant results with MR-egger and MR-PRESSO (p-value > 0.05). When investigating the opposite direction of causality (i.e. insulin resistance -> obesity) using a similar pipeline, we found no genome-wide significant SNPs associated with the Matsuda index in METSIM or other cohorts of previous studies. Furthermore, no insulin resistance parameters are measured in the UKB. Therefore, the first assumption of MR (i.e. IV is associated with the exposure variable) cannot be fulfilled. As METSIM may be underpowered to identify genome-wide significant SNPs for the Matsuda index, the current sample size of the Matsuda index GWAS does not allow a reliable MR analysis in this opposite direction. Accordingly, assessment of this direction using MR warrants further investigation in larger GWAS cohorts with the Matsuda index available for study.

To investigate whether potential confounders affect our MR analyses, we tested for association between the IVs and 3 potential confounders, i.e. alcohol intake, smoking, and physical activity. S1 Fig shows that in all 3 MR analyses, only 4 IV variants (rs7187776, rs1788783,

rs1872841, and rs62037364) showed a significant association with these confounders (S1A Fig), and removing these 4 variants did not affect our MR results (S1B and S1C Fig), and thus obesity still showed a significant causal effect on both insulin resistance and prediabetes.

In summary, we established a one-way causal effect of obesity on prediabetes among the non-diabetic individuals from the UKB. Although MR-egger did not identify a similar significant causal effect of body fat percent on prediabetes as MR-PRESSO, the pleiotropy test employed by MR-egger and a heterogeneity test did not identify evidence for pleiotropy, which suggests no violation of the no-pleiotropy assumptions required by an MR analyses. In contrast, prediabetes did not show any evidence for a causal effect on body fat percent. We then followed up this finding by identifying a negative causal effect of body fat percent on insulin sensitivity (i.e. Matsuda index) in the non-diabetic individuals from METSIM. Although METSIM is underpowered to investigate the reverse-causal effect (i.e. insulin resistance -> body fat percent), the MR analyses performed in both UKB and METSIM suggest that obesity causes prediabetes and insulin resistance before the development of diabetes and thus, prediabetes is less likely to cause obesity among the non-diabetic population.

## Adipose mitochondrial (MT) gene expression plays a key role in insulin resistance

We have shown that obesity (i.e. a high body fat percent) leads to increased insulin resistance in human using MR analysis. Since obesity reflects the amount of body fat, adipose tissue may play an important role in obesity-induced insulin resistance. Thus, we further investigated how adipose tissue affects the Matsuda index in the METSIM cohort. Among the 4k unrelated individuals in METSIM, 335 had bulk RNA-seq data from the subcutaneous adipose tissue biopsies. To estimate MT gene expression, we estimated the transcripts per million (TPM) values of each gene in the RNA-seq data and used the sum of TPMs from all 37 MT encoded genes to represent the MT gene expression. We also included the first 3 genetic PCs as covariates when correcting the MT expression to adjust for the potential population stratification (see Methods). We first observed that MT expression is significantly associated with BMI and body fat percent in the Finnish METSIM cohort using a threshold of p-value < 0.05 (S2A and S2B Fig). We also verified this association between the obesity stage and MT expression in another population, and our data in the Mexican cohort (see Methods) show that the lean Mexicans have significantly higher MT adipose expression when compared to the obese Mexicans (p-value = 0.016) (S2C Fig).

Next, we corrected the Matsuda index in METSIM for age, age$^2$ and excluded people who have type 2 diabetes (T2D) (n = 11). The Matsuda index and MT gene expression were inverse normal transformed to obtain normal distribution. Using these data, we discovered that the MT gene expression is significantly associated with Matsuda index (p-value = $9.60 \times 10^{-15}$, n = 324) in METSIM (Fig 2A). To further replicate and validate this finding, we tested the association between MT gene expression and insulin resistance in the RNA-seq data from 5 different tissues in the GTEx cohort. Since the GTEx cohort does not have the Matsuda index measured, we tested the MT gene expression difference between the patients with T2D and non-diabetic individuals. Fig 2B shows that, as in METSIM, the patients with T2D in GTEx have significantly lower MT gene expression in the subcutaneous and visceral adipose tissue than the non-diabetic individuals (S3A Fig). However, in three other non-adipose tissues from GTEx, only muscle MT gene expression (n = 305) showed the significant difference between T2D patients and non-diabetics (P = $3.42 \times 10^{-2}$). In the liver and whole blood (total n = 412), no significant difference was observed using the threshold of p-value < 0.05 (S3 Fig). These results support the important role of adipose tissue and muscle in the insulin resistance related

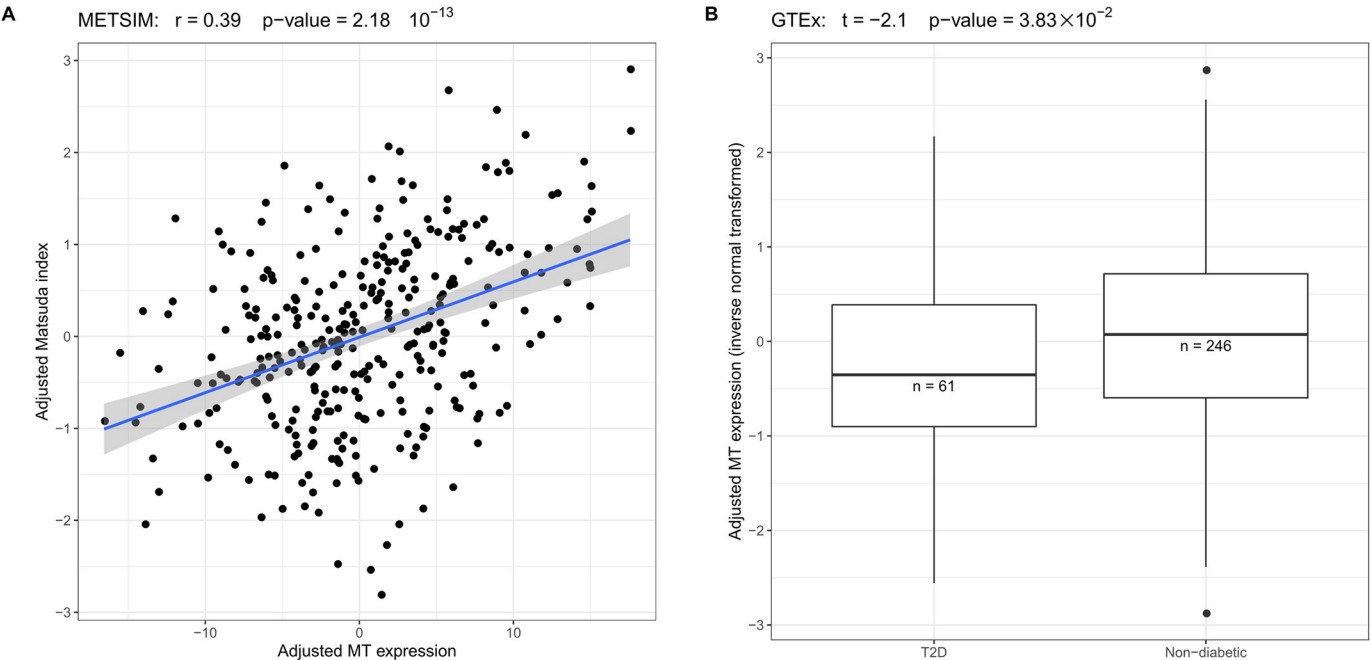

**Fig 2. A low adipose MT gene expression is associated with insulin resistance.** (A) In METSIM, the adjusted adipose MT gene expression is significantly associated with the adjusted Matsuda index. (B) In GTEx, the non-diabetic individuals have significantly higher adjusted MT gene expression (inverse normal transformed) than the T2D patients.

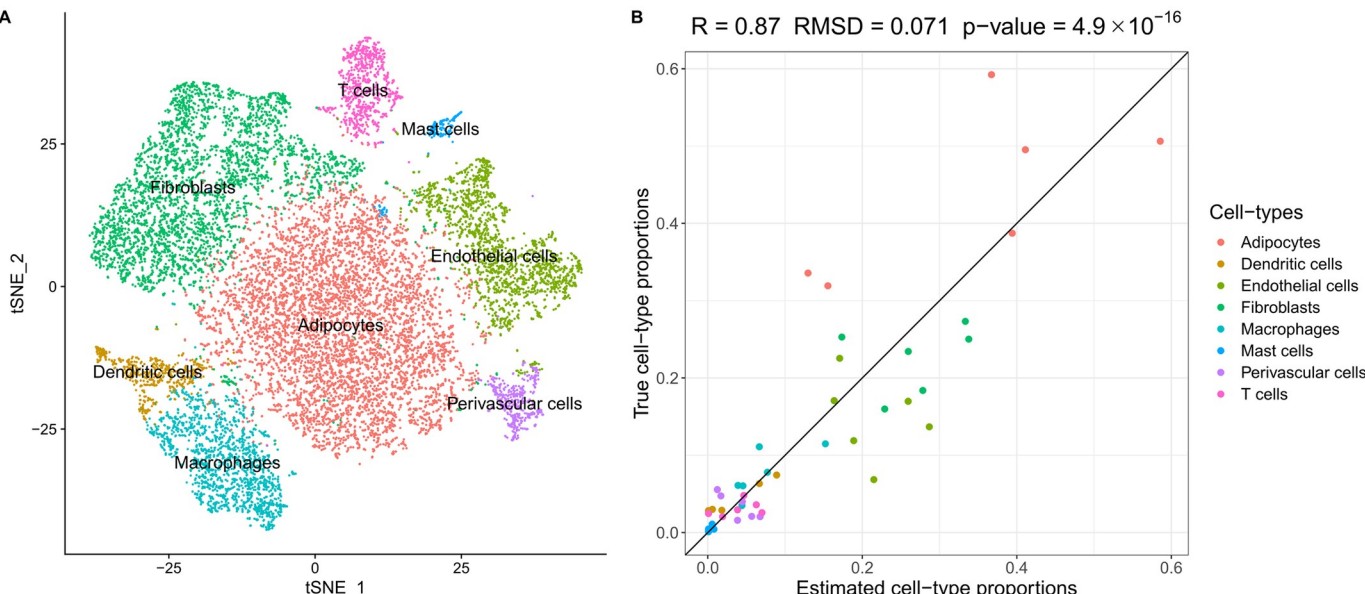

**Fig 3. Analysis of sn-RNA-seq data reveals tissue heterogeneity in human subcutaneous adipose tissue.** (A) We identified 8 cell-type clusters in 15,623 nuclei from frozen human adipose tissue from 6 Finnish individuals. The t-SNP plot is colored by the identified cell types. (B) Using sn-RNA-seq as reference, the estimated adipose cell-type proportions from bulk adipose RNA-seq data are well concordant with the true cell-type proportions.

metabolic process; however, due to the limited sample size of the GTEx liver cohort (n = 123), we cannot exclude the potential role of liver in insulin resistance. Taken together, the adipose MT gene expression is significantly associated with insulin resistance in the METSIM cohort and T2D in the GTEx cohort. In both subcutaneous and visceral adipose tissue, the MT gene expression is significantly lower in insulin resistant individuals.

## Assessing tissue heterogeneity and adipose cell type proportions using single nucleus RNA sequencing

**Single nucleus RNA sequencing reveals 8 cell-types in human adipose tissue.** Adipose tissue is a complex tissue that consists of multiple cell-types, such as adipocytes, preadipocytes, macrophages, fibroblasts, and vascular cells. Even though adipocytes comprise ~90% of the total volume in the human adipose tissue, they only take ~50% of the total cell count [45–47]. We hypothesized that the metabolic processes in different contexts and adipose cell-types associated with obesity may substantially affect systemic insulin resistance. To investigate how adipose cell-type heterogeneity affects insulin resistance, we performed single-nucleus RNA sequencing (sn-RNA-seq) on 6 frozen human subcutaneous adipose tissue biopsies (see the Methods) and used cell-type-specific gene expression data as a reference to identify signature genes for each adipose cell-type in order to estimate cell-type proportions from the bulk adipose RNA-seq profiles.

Using the 10x Genomics platform, we sequenced on average ~2,600 nuclei for each sample and obtained the non-zero expression of ~500 genes per cell (for sample-specific metrics, see S1 Table). Next, we used Seurat to cluster the sn-RNA-seq data and identified 8 adipose cell-type clusters based on the gene expression profiles of the adipose nuclei. It is worth noting that the adipocyte cluster comprises 44.0% of the total cell number, which is in line with the previous findings [47]. Fig 3A shows the tSNE plots of the 8 adipose cell-type clusters in 15,623 nuclei. S4A Fig shows the tSNE plot that is colored by the sample IDs. These data show that clustering is largely driven by distinct gene expression profiles from different adipose cell-types rather than by the differences between individuals.

**Estimating adipose cell-type proportions using the sn-RNA-seq as the reference.** Next, we used MuSiC to estimate the proportions of each cell-type from the bulk adipose RNA-seq data. Utilizing both bulk RNA-seq and sn-RNA-seq data from these 6 individuals, we estimated the cell-type proportions of the 6 individuals from bulk RNA-seq data and compared the decomposition results with the true cell-type proportions from the sn-RNA-seq data to verify our decomposition method. We employed a leave-one-out approach to decompose the cell-type proportions of each sample while using the sn-RNA-seq data of the other 5 samples as the reference. Fig 3B shows that the estimated adipose cell-type proportions have a high concordance with the true adipose cell-type proportions. Thus, our decomposition method provides reliable estimated adipose cell-type proportions then we used the 6 sn-RNA-seq samples as reference. S2 Table lists the true cell-type proportions of the 6 sn-RNA-seq samples.

After verifying the accuracy of cell-type decomposition by MuSiC, we applied the method to the 335 subcutaneous adipose bulk RNA-seq samples from the METSIM cohort to estimate the proportions of the 8 adipose cell-types (S3 Table). We first checked the associations between the Matsuda index and the 8 estimated cell-type proportions using linear regression. In the association tests, we inverse normal transformed the Matsuda index and included age as a covariant. Four of 8 estimated cell-type proportions showed a significant association (Bonferroni adjusted p-value < 0.05) with the Matsuda index (S4 Table). It is worth noting that both dendritic cells and macrophages exhibited a strong association with the Matsuda index (p-

values $< 5 \times 10^{-5}$), suggesting that immune cell types contribute to insulin resistance in human adipose tissue.

**Body fat percent and adipose MT gene expression are independently associated with systemic insulin resistance after adjusting for tissue heterogeneity.** It has been shown previously that a high BMI associates with adipose MT activity [48]. In line with this we observed that MT gene expression is significantly associated with body fat percent (p-value = $3.55 \times 10^{-7}$) in the adipose RNA-seq data from METSIM (n = 324). However, it is unknown if body fat percent or adipose tissue heterogeneity causes the association between the adipose MT expression and insulin resistance. To investigate this, we explored the associations between the Matsuda index and age, body fat percent, adipose MT gene expression, and adipose cell-type proportions using a multi-variable linear model (see model 1 in Methods). In our multi-variable linear model, body fat percent, MT expression, and the estimated proportions of dendritic and fibroblasts cells in adipose all showed significant associations with the Matsuda index (p<0.05) (S5 Table). This result suggests that the adipose tissue heterogeneity, MT gene expression, and body fat percent all have independent contributions to the variance in the Matsuda index. Noteworthy, this model explained 43.37% of the variance ($R^2$) in the Matsuda index, which is higher than using any trait alone: body fat percent ($R^2$ = 30.89%); MT expression ($R^2$ = 14.64%); and estimated cell-type proportions ($R^2$ = 29.24%). Moreover, when we excluded body fat percent from model 1, the estimated adipose cell-type proportions and MT expression together explained a substantial amount ($R^2$ = 35.42%) of the variance in the Matsuda index. The high variance explained by model 1 makes it possible to predict the Matsuda index, i.e. systemic insulin resistance, using body fat percent, adipose MT gene expression and the estimated adipose cell-type proportions.

## Utilizing adipose RNA-seq data to predict systemic insulin resistance

Although the Matsuda index is an important biomarker for glucose metabolism, its measurement requires an oral glucose tolerance test, which is not available in many adipose RNA-seq cohorts. To this end, we developed a prediction model using elastic net regularization [49] that combines body fat percent/BMI, MT gene expression, age, and cell-type proportion information to predict the Matsuda index using adipose RNA-seq data. Although body fat percent is a more ideal trait for obesity compared to BMI, BMI is easy to measure, which makes it widely available in most human adipose RNA-seq cohorts. Thus, we trained two models that use either BMI or body fat percent to present obesity status to predict Matsuda index in METSIM. Employing a 100-fold cross validation in both prediction models, S5 Fig shows that the predicted Matsuda index has a high concordance with the true Matsuda index (r > = 0.64, p-value < = $1.65 \times 10^{-38}$) in both models. Since body fat percent and BMI exhibited a similar accuracy in predicting Matsuda index, we further tested the BMI model, which can potentially be applied to more cohorts. To further confirm this promising prediction of insulin resistance, we used the METSIM cohort to train a model and then estimated the Matsuda index in two independent adipose RNA-seq cohorts: GTEx and FTC (see Methods). Fig 4B shows that in the GTEx subcutaneous adipose samples, the T2D patients have significant lower predicted Matsuda index when compared to the non-diabetic GTEx individuals (p-value = $2.62 \times 10^{-5}$). Since the monozygotic twin participants share the identical genetic background, we tested the association between the predicted Matsuda index and the true Matsuda index both in the full FTC cohort and the unrelated individuals by randomly selecting one individual from each twin pair. Fig 4C and 4D shows that the predicted Matsuda index is similarly well concordant with the true Matsuda index in the full FTC cohort (r = 0.51, p-value = $1.15 \times 10^{-7}$) and in the unrelated FTC individuals (r = 0.46, p-value = $1.03 \times 10^{-3}$).

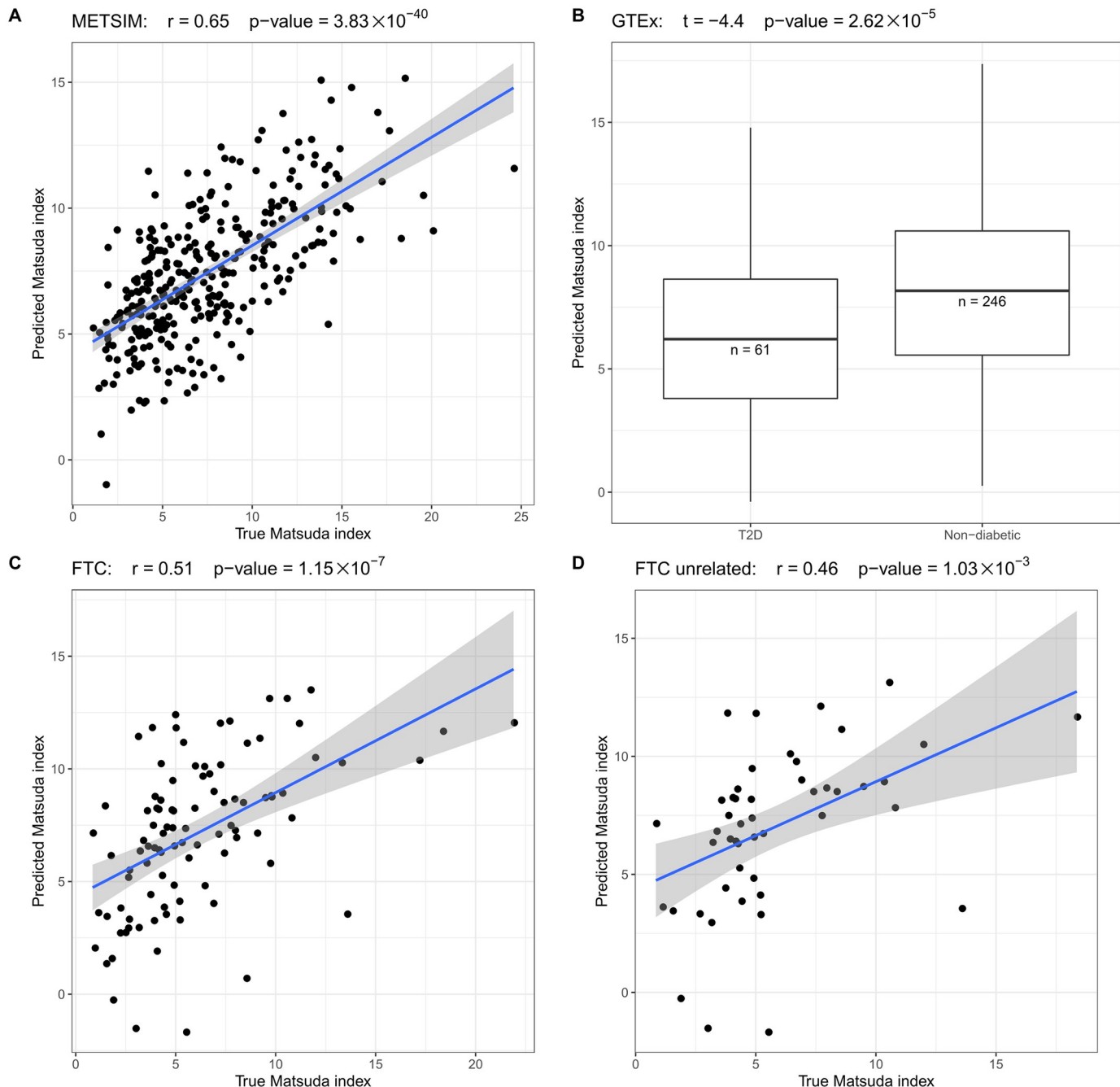

**Fig 4. The predicted Matsuda index is well concordant with the true Matsuda index values in 3 different cohorts: METSIM, GTEx, and FTC.** (A) In METSIM, the estimated and true Matsuda index are significantly associated. (B) In GTEx, the predicted Matsuda index is significantly higher in non-diabetic individuals when compared to the T2D patients. (C) In FTC, the estimated and true Matsuda index are significantly associated. (D) We randomly choose one individual from each twin pair to select the unrelated individuals from FTC. Among the unrelated individuals, the estimated and true Matsuda index are also significantly associated, indicating that the twin status did not bias the prediction of the Matsuda index.

Furthermore, the predicted Matsuda index had the best concordance with the true Matsuda index when compared to any of the tested traits alone in METSIM and FTC. S6 Fig shows that neither MT gene expression nor BMI can predict the Matsuda index as accurately as our prediction model in the METSIM and FTC cohorts. Therefore, this prediction model can

potentially be used to impute systemic insulin resistance into other adipose RNA-seq cohorts, in which this key glucose metabolism trait has not been measured. S6 Table shows the betas in the trained prediction model that can be applied to other cohorts. In summary, we discovered that a substantial amount (44.39%) of the variance in the systemic insulin resistance, measured using the Matsuda index, can be explained by adipose MT gene expression, adipose cell-type proportions, and BMI. By combining the information from these traits, we were repeatedly able to predict the Matsuda index with a great accuracy when compared to the prediction results with any single trait alone. Since the Matsuda index is an important biomarker for glucose metabolism in humans, our prediction model can be utilized to impute the Matsuda index into adipose RNA-seq cohorts where this key metabolic trait is missing.

## Discussion

Even though the previous MR studies have shown that obesity has a causal effect on T2D [50,51], MR evidence showing that obesity is causally linked with prediabetes or insulin resistance among non-diabetic individuals is missing. Noteworthy, the annualized conversion rate of prediabetes to diabetes is estimated to be 5%–10% [9] and thus, some prediabetes patients do not develop T2D. Thus, the causal effect between obesity and prediabetes cannot be simply deduced by the causal role of obesity on T2D. In the present study, we utilized the extensive UK Biobank cohort [20] and carefully phenotyped Finnish METSIM cohort [21] to investigate whether obesity causes prediabetes and causally increases insulin resistance in the non-diabetic population using the MR analysis. Our MR result sheds new light on the long-standing reverse causality question between obesity and insulin resistance by establishing its directionality. Stancakova et al. have showed earlier that the Matsuda index is the best index of insulin sensitivity when compared to other surrogate indexes of insulin resistance using an M value from the euglycemic hyperinsulinemic clamp as the gold standard [52]. Therefore, the Matsuda index is largely a measure of systemic rather than adipose-based insulin resistance. However, when we examined the role of adipose cell-type heterogeneity, adipose MT gene expression, and BMI in systemic insulin resistance, we discovered that even when excluding BMI from the calculation, the estimated adipose cell-type proportions and adipose MT gene expression together still explain a substantial amount ($R^2 = 35.42\%$) of the variance in the Matsuda index. When we included BMI into this analysis, all three factors are independently associated with insulin resistance ($p<0.05$ for each) and the $R^2$ increased to 44.39% which is higher than using any trait alone ($R^2 < = 30.89\%$). This surprisingly high proportion of variance explained by adipose tissue (i.e. adipose cell types and MT gene expression) and BMI suggests that adipose tissue has an important role in the systemic insulin resistance. Based on this novel finding, we built a prediction model using adipose cell-types, adipose MT gene expression, and BMI that accurately predicted insulin resistance across multiple cohorts.

To investigate how adipose tissue heterogeneity affects systemic insulin resistance, we performed sn-RNA-seq using 6 frozen human subcutaneous adipose tissue samples. Noteworthy, the previous studies investigating adipose cell-type heterogeneity used FACS [25–27]; however this application is limited to a small number of well identified non-adipocyte cell-types and is unable to detect refined new subtypes. There are no previous publications performing sn-RNA-seq from frozen human adipose tissue, and thus the role of cell-type heterogeneity in insulin resistance has not been investigated for all main adipose cell types before. After careful quality control, the sn-RNA-seq generated 15,623 nuclei from the 6 adipose tissue biopsies and identified 8 adipose cell-type clusters based on their gene expression. We then used the sn-RNA-seq data as the reference data to detect cell-type specific signature genes in each adipose cell type cluster and decomposed the cell-type proportions in the METSIM, FTC, and GTEx

adipose bulk RNA-seq cohorts, leveraging thus substantially the information contained in these existing bulk RNA-seq cohorts. Notably, the estimated cell-type proportions of macrophages and dendritic cells exhibited a significant association with insulin resistance, demonstrating the key role of the obesity-induced low-grade inflammation process in systemic insulin resistance [10,11,13–15].

Even though insulin resistance is an essential clinical metabolic trait in obesity-related cardiometabolic diseases, it is often not measured in the existing adipose RNA-seq cohorts, such as the GTEx cohort [55]. Moreover, although the adipose tissue is suggested to be relevant in the development of insulin resistance [12,16,34], to the best of our knowledge, the variance in insulin resistance parameters that can be explained by adipose tissue has not been reported previously. Strikingly, we found that 44.39% of the variance in systemic insulin resistance (i.e. the Matsuda index) can be explained by the adipose cell-types, adipose MT expression, and BMI using the METSIM cohort. Thus, we developed an elastic net prediction model to predict the Matsuda index using these traits. The prediction model was trained in a subset of the METSIM cohort. The model not only successfully predicted the Matsuda index in the METSIM test cohort but also predicted well the Matsuda index in 2 independent cohorts, the GTEx and FTC, indicating that we can predict the missing systemic insulin resistance estimates to cohorts lacking metabolic phenotype data, such as GTEx. Since the prediction model is based on adipose RNA-seq data, the predicted Matsuda index can potentially be used to proportionally estimate how much of insulin resistance is driven by adipose tissue versus other metabolic tissues. This would help subtype different forms of insulin resistance states underlying the development of type 2 diabetes.

As we used RNA-seq data to predict the cell-type proportions and then utilized those for the prediction of the Matsuda index, this two-step prediction design may introduce noise from both steps. However, utilizing gene expression across different cohorts is known to be prone to technical biases and batch effects [53], whereas the estimated cell-type proportions represent more uniform traits when compared to the gene expression. Moreover, as we have shown that the estimated cell-type proportions have relatively high associations with the true cell-type proportion, the cell-type proportions estimated by other methods, such as FACS sorting, can potentially also be used to predict Matsuda index in other studies.

In summary, we have shown that obesity has a strong participation in prediabetes and insulin resistance using the MR analysis. By leveraging bulk RNA-seq data in large adipose RNA-seq cohorts using a small amount of adipose sn-RNA-seq data to decompose adipose cell-types, we show that a substantial proportion (44.39%) of systemic insulin resistance can be explained by certain adipose cell-type proportions, MT gene expression, and BMI. This new finding not only establishes the key role of adipose tissue in regulating insulin resistance but also provides a useful method to impute insulin resistance estimates to human transcriptome cohorts.

## Materials and methods

### Ethics statement

The METSIM, MOSS and FTC study designs were approved by local ethics committees and all participants gave a written informed consent.

### Study cohorts

We analyzed the genotype and phenotype data of ~510k individuals from two cohorts: METSIM cohort (n = 10,198) [21], and UK Biobank cohort [20] (n = 391,816) In the METSIM cohort, middle-aged Finnish males were recruited at the University of Eastern Finland and

Kuopio University Hospital, Kuopio, Finland, and the biochemical lipid, glucose, and other clinical and metabolic phenotypes were measured as described previously [21]. Briefly, a 2-h oral glucose tolerance test (OGTT) (75 g of glucose) was performed in the METSIM cohort, and samples for plasma glucose and insulin were drawn at 0, 30, and 120 min [21]. We evaluated insulin resistance in the non-related, non-diabetic METSIM participants using the Matsuda index that was calculated based on the OGTT values, as described in detail previously [54]. The METSIM study design was approved by local ethics committee and all participants gave a written informed consent. This research has been conducted using the UK Biobank Resource under Application Number 33934. The UK Biobank data was downloaded from the UK Biobank data repository on 08/23/2018.

We analyzed the RNA-seq data in 751 human subcutaneous adipose samples from 4 different cohorts: METSIM cohort (n = 335) [21], Genotype-Tissue Expression (GTEx) cohort (n = 308) [55,56], the ongoing MOSS cohort (n = 107), and 54 monozygotic Finnish monozygotic twin cohort (FTC) (n = 108) [31,57,58]. The METSIM, MOSS and FTC study designs were approved by local ethics committees and all participants gave a written informed consent. The GTEx adipose RNA-seq data were downloaded from dbGaP (accession number phs000424.v6.p1) on 08/11/2016. In addition to subcutaneous adipose tissue, we also analyzed the GTEx visceral adipose tissue, blood, liver, and muscle RNA-seq data (v7). In the MOSS cohort, we collected subcutaneous adipose tissue biopsies from extreme obese Mexican individuals undergoing bariatric surgery in Instituto Nacional de Ciencias Medicas y Nutricion (INCMN), Mexico City. Currently, we have RNA-seq data from subcutaneous adipose samples from 88 obese Mexican MOSS participants and 19 lean Mexicans (BMI < 25) as controls. We used the FTC cohort to verify the prediction model of the Matsuda index. In this cohort, we generated RNA-seq data from subcutaneous adipose tissue of 54 MZ twin pairs (n = 108). S7 Table shows the clinical characteristics of the participants in the 4 cohorts. We also selected the adipose biopsies of 6 individuals from FTC for the sn-RNA-seq experiment. S1 Table shows the phenotypic characteristics of the 6 Finnish individuals whose adipose biopsies were processed for sn-RNA-seq. The 6 individuals have roughly similar ages, 3 of the 6 are males, and 3 of the 6 have a normal BMI (BMI<25).

## GWAS and Mendelian randomization (MR) analysis

To identify candidate instrumental variables (IVs) for the two-sample MR analysis in UKB, we first randomly separated the UKB cohort into two independent groups and performed GWAS analyses of body fat percent and prediabetes in different groups. The prediabetes cases were identified by serum HbA1c level between 5.7–6.4 [38]. We excluded the individuals who had HbA1c > 6.4 or had been diagnosed as diabetic to ensure that only non-diabetic individuals were included in the GWAS analyses. We used BOLT-LMM [59] to explore the associations between the genotypes and the target phenotype, while accounting for the population stratification. We inverse normal transformed body fat percent to ensure a normal distribution and included age, $age^2$, sex, array type, center ID, and 20 genotype PCs as covariates. The percent of prediabetes cases in UKB is 14% (62,866 out of 441,057), which is in line with the BOLT-LMM recommendation of = > 10% case fraction for an unbiased case-control GWAS study [60]. To fulfill the first assumption of MR (i.e. IVs should be significantly associated with the exposure variable), we selected the independent (R2<0.01) GWAS SNPs (p-value<5e-8 in UKB) of body fat percent and prediabetes as candidate IVs for the MR analysis. Then we used MR-PRESSO [43] to identify causal associations between body fat percent and prediabetes while controlling for the potential pleiotropy. We also used MR-egger [61] and heterogeneity

test (Cochran's Q test) to verify that no pleiotropy biased our MR analysis. S8 Table shows the summary statistics employed in the MR analyses.

When searching for a causal effect of body fat percent on insulin resistance, we performed GWAS analyses of body fat percent and Matsuda index using the non-diabetic individuals METSIM using BOLT-LMM [59] following the same pipeline as in UKB. For these GWAS analyses, we inverse normal transformed body fat percent and Matsuda index and included age, age$^2$, and 10 genotype PCs as the covariates. In the GWAS, we did not identify any Matsuda index-associated SNPs in METSIM using a genome-wide significant cut point of p-value$<5.0\times10^{-8}$. For body fat percent, we performed a GWAS analysis on body fat percent in UKB using BOLT-LMM and utilized the non-redundant SNPs ($R^2<0.01$) as the candidate IVs (n = 241) (S9 Table). Since UKB has a larger sample size than METSIM, the IVs identified in UKB do not have genome-wide significant associations (p-value$<5.0\times10^{-8}$) with body fat percent in METSIM. We then utilized MR-PRESSO to identify potential pleiotropy and test for the causal effect between body fat percent and the Matsuda index in the direction obesity -> insulin resistance. Noteworthy, there is no theoretical "perfect" MR method that guarantees a pleiotropy free MR analysis. Thus, the absence of evidence for pleiotropy does not necessarily prove the absence of pleiotropy. S9 Table shows the summary statistics employed in the MR analyses. The opposite causal direction, insulin resistance -> obesity, could not reliably be assessed using IVs due to the lack of genome-wide significant Matsuda index SNPs in the METSIM cohort (see Results for details).

## Single nucleus RNA-sequencing and clustering

Frozen subcutaneous adipose tissue was minced over dry ice and transferred into ice cold lysis buffer consisting of 0.1% NP-40, 10mM Tris-Hcl, 10 mM NaCl, and 3 mM MgCl2. After a 10-minute incubation period, the lysate was gently homogenized using a dounce and filtered through a 70 μm MACS smart strainer (Miltenyi Biotec #130-098-462) to remove debris. Nuclei were centrifuged at 500 g for 5 minutes at 4°C and re-suspended in wash buffer consisting of 1X PBS, 1.0% BSA, and 0.2 U/μl RNase inhibitor. We further filtered nuclei using a 40 μm Flowmi cell strainer (Sigma Aldrich # BAH136800040) and centrifuged at 500 g for 5 minutes at 4°C. Pelleted nuclei were re-suspended in wash buffer and immediately processed with the 10X Chromium platform following the Single Cell 3′ v2 protocol.

We used Cell Ranger [62] to build a pre-mRNA alignment reference based on the reference gencode 19 and estimate the UMIs in each cell. As the quality control, we excluded the cells that had <300 genes expressed and kept only the genes that were expressed in at least 3 cells. Then we used Seurat [63] to simultaneously cluster all the qualified cells from the 6 individuals. We identified 8 clusters and 697 signature genes (S10 Table) that have a higher expression in one of the clusters over the others.

## Decomposition of adipose cell-type proportions from bulk RNA-seq data

We first used Cell Ranger to re-align the single nucleus reads to a mature mRNA reference (gencode 19) and then estimated the pseudo-bulk gene expression in the 6 individuals. Next, treating the candidate gene expression of the sn-RNA-seq data as the reference, we used MuSiC [64] to estimate the cell-type proportions from the bulk RNA-seq data. To validate the accuracy of our decomposition method, we performed both sn-RNA-seq and bulk RNA-seq using the subcutaneous adipose biopsies from the same 6 individuals. Then we predicted the cell-type proportions using the bulk RNA-seq data and compared the decomposition results to the cell-type proportions estimated from the sn-RNA-seq data of the same individuals. To ensure the independence of the test data, we used the leave-one-out strategy. In more detail,

when we estimated cell-type proportions of one individual, we used the sn-RNA-seq data from the other 5 individuals as the reference. Thus, all of the estimated cell-type proportions of each individual are based on an unrelated data set.

Because sn-RNA-seq captures not only mature mRNAs but also pre-mRNAs, the expression patterns of some genes are expected to be different between the nuclei and bulk RNA-seq data. For example, the MALAT1 gene (ENSG00000251562) exhibits an average TPM of 254 in the bulk adipose tissue in METSIM while its average TPM in the sn-RNA-seq data is 391,375. Accordingly, we observed that the decomposition results were biased by these different expression patterns when we used all the ~16,000 expressed genes as suggested by MuSiC (S4B Fig). To improve the accuracy of the decomposition, we calculated the difference in mean of the log-transformed gene expression across all genes from the target bulk RNA-seq samples and the 6 pseudo-bulk samples. Then we normalized the expression differences and kept the genes that have chi square statistic <= 1. After this filtering process, we kept ~4,000 genes that have similar expression in both the single nucleus and bulk RNA-seq data. When estimating the cell-type proportions in the bulk RNA-seq data, we used the sn-RNA-seq data from all of the 6 samples as the reference. Since cell-type proportions are estimated from RNA-seq data that is affected by technical factors, we also adjusted the proportion of each cell-type for RNA-seq technical factors when testing for the association between cell-type proportions and traits in the METSIM cohort.

## QC for estimating MT gene expression

We used the same pipeline to estimate MT expression in all cohorts. First, we used FastQC [65] to verify the sequence quality of the RNA-seq data. Then, we performed a 2-pass alignment using STAR [66] (reference genome: gencode 19, hg19) and subsequently used feature-Counts [67] to estimate the TPM of each gene. Only uniquely mapped reads were counted for gene expression. MT gene expression was defined as the sum of TPMs of all MT encoded genes. Since gene expression estimates from RNA-seq data are affected by multiple technical factors [68], we corrected MT gene expression for 11 known technical factors (S11 Table) and 3 genotype PCs. We chose to correct for 3 genotype PCs to follow a similar pipeline as implemented in the GTEx project [56]. Since the GTEx RNA-seq samples were collected from deceased individuals, we also adjusted MT gene expression for the post-mortem sample collection time in the GTEx cohort.

To control for admixed ancestry in the MOSS cohort, we called variants from the RNA-seq data following the GATK pipeline. We used the recommended parameters of -window 35, -cluster 3, and filtering FS> 30 and QD <2. Furthermore, we used variants with MAF >5% and an average read depth > = 30. We then combined the MOSS and 1000 Genomes Project genotype data [69] and performed PCA, observing that the MOSS individuals clustered well with the individuals of Amerindian descent. Accordingly, we used these SNPs called from RNA-seq data to calculate the genotype PCs in the MOSS cohort for the correction of ancestry.

However, it is worth noting that the MT genome is small and has a simple structure when compared to the autosomal chromosomes. Thus, RNA metrics estimated from MT reads have a different pattern compared to that of autosomal reads. Since MT reads comprise a relatively large proportion of total reads (S11 Table), we discovered that the technical factors estimated from the RNA-seq data, such as intergenic read percent and exonic read percent, are heavily correlated with the MT read percent of each sample. S7A and S7B Fig shows that in the MET-SIM cohort, almost all the RNA metrics estimated by Picard Tools [70] show a significant association with the MT read percent, with the percent of intergenic reads exhibiting the strongest association with the MT read percent (R = 0.86, p-value = 7.54 x $10^{-103}$). Since the MT read percent reflects MT gene expression, these correlated RNA metrics cannot well represent the

true technical covariation. Correcting for these factors when estimating MT gene expression would thus remove signals from MT gene expression. To address this issue, we first excluded the MT reads from the RNA-seq data and then estimated these technical factors from the reads aligned to the nuclear genome using Picard Tools. The new unbiased technical factors showed much weaker associations with the MT read percent (S7C and S7D Fig).

## Disentangling the associations between MT expression, body fat percent, insulin resistance (i.e. the Matsuda index), and tissue heterogeneity

Since individuals with T2D are insulin resistant and the antidiabetic medication may influence the outcome, we removed them from all analyses involving the Matsuda index. We built a multi-variable linear model treating the Matsuda index as the dependent variable to identify the associations between MT expression, BMI/body fat percent, estimated adipose cell-type proportions and Matsuda index in the METSIM cohort:

$$Matsuda \sim \beta_b * body\ fat\ percent + \sum \beta_{ci} * CT_i + \beta_M * MT \qquad \text{(Model1)}$$

MT indicates the corrected MT gene expression. Matsuda indicates the Matsuda index. $CT_i$ is the estimated cell-type proportion in adipose tissue. The $\beta$s are the estimated parameters from the multi-variable linear models. Since the sum of the 8 estimated cell-type proportions equals to 1, the degree of freedom of the cell-type proportions is 7 instead of 8. We excluded the proportion of endothelial cells from the model due to its less accurate prediction when compared to the other cell-types.

To predict the Matsuda index using the other traits, we employed the following elastic net regularization [71] to select the predictors and predict the β for each variable in model 1:

$$\hat{\beta} = \underset{\beta}{\operatorname{argmin}}(\|Matsuda - X\beta\| + \lambda\|\beta\| + \lambda\|\beta\|^2)$$

Elastic net regularization applies a penalty on the $\beta$s estimated from the predictors, and therefore reduces the estimated $\beta$s of unnecessary predictors to 0. Thus, the predictors with a non-zero estimated $\beta$ are kept in the prediction model. We used the 'glmnet' package [72] to obtain the $\lambda$ that has the minimum mean cross-validated error in the training data set, and then used the specified $\lambda$ and $\beta$s to predict the Matsuda index. To evaluate the prediction accuracy in both models, we performed a 100-fold verification in the METSIM cohort. In more detail, we randomly split the individuals into 100 groups, and then for each group, we predicted its value based on the model that we trained with the other 99 groups. We also verified this model in 2 independent cohorts: FTC and GTEx. For building the final prediction model for these 2 cohorts, we used all individuals in the METSIM cohort as the training set and predicted the Matsuda index in GTEx and FTC as verification. Using the predicted Matsuda index, we performed a Pearson correlation test to check the association between the estimated and true Matsuda index. Since GTEx do not have the Matsuda index available, we compared the predicted Matsuda index values between the GTEx individuals with and without T2D as the verification.

## Supporting information

**S1 Fig. Removing the IVs associated with 4 potential confounders did not affect the MR results.** (A) The associations between all IVs and three potential confounders, i.e. alcohol intake, physical activity, and smoking. The x axis shows the MR analysis in which the IVs were used. The y axis shows the p-values of the association between the IVs and confounders. (B) The MR results of body fat percent on Matsuda index without using the confounder-associated variants as IVs.

(C) The MR results of body fat percent on prediabetes without using the confounder-associated variants as IVs. (D) The MR results of prediabetes on body fat percent remained the same as in Fig 1D as none of the prediabetes IV SNPs were associated with the 3 tested confounders.
(TIF)

**S2 Fig. In METSIM and MOSS, we verified the significantly lower MT expression (p-value < 0.05) in the obese individuals who have a high body fat percent or BMI compared to the lean individuals.** (A) BMI is significantly associated with the adjusted MT expression in a Pearson correlation test. (B) Body fat percent is significantly associated with the adjusted MT expression in a Pearson correlation test. (C) In MOSS, obese Mexicans have a lower adjusted MT expression when compared to the lean Mexicans.
(TIF)

**S3 Fig. In GTEx, we used a t-test to show the association between the adjusted MT gene expression (inverse normal transformed) and T2D status in 4 different tissues.** In (A) visceral adipose and (B) muscle, the adjusted MT expression is significantly higher in the non-diabetic individuals than in the T2D patients. In (C) liver and (D) blood, there is no evidence of differential MT gene expression between the non-diabetic individuals and the T2D patients.
(TIF)

**S4 Fig. Our QC process ensures that the SN-RNA-seq accurately estimated the cell-type proportions in subcutaneous adipose tissue.** (A) The t-SNE plot shows no evidence of a batch effect in SN-RNA-seq clustering. The dots are colored by sample IDs. (B) When using all genes in the SN-RNA-seq data without any filtering, the estimated cell-type proportions are not concordant with the true cell-type proportions. Thus, using the selected genes (Fig 2B) performs much better than using all genes as reference in the decomposition process.
(TIF)

**S5 Fig. Person correlations show that BMI and body fat percent are highly correlated in the METSIM and FTC cohorts.** Using either BMI or body fat percent predicted Matsuda index in METSIM cohort in a similar accuracy. (A) The association between the body fat percent and BMI in METSIM. (B) The association between the body fat percent and BMI in FTC. (C) The association between the predicted Matsuda index and true Matsuda index when using BMI to represent obesity status. (D) The association between the predicted Matsuda index and true Matsuda index when using body fat percent to represent obesity status.
(TIF)

**S6 Fig. Pearson correlations show the associations between the predictors (BMI, corrected MT expression) and Matsuda index in the METSIM and FTC cohorts.** The predicted Matsuda index is always more strongly associated with the true Matsuda index than any of the predictors (Fig 4). (A) The correlation between raw BMI and the Matsuda index in METSIM. (B) The correlation between the corrected MT gene expression and Matsuda index in METSIM. (C) The correlation between raw BMI and the Matsuda index in FTC. (D) The correlation between the corrected MT gene expression and Matsuda index in FTC.
(TIF)

**S7 Fig. Using the METSIM cohort as an example, we demonstrate that when estimating RNA metrics, including the MT reads, the RNA metrics are heavily biased by the MT read percent.** When excluding the MT reads to estimate the RNA metrics, the correlations between the MT read percent and other RNA metrics are reduced. (A). The qq-plot shows the correlations between the MT read percent and estimated RNA metrics, including MT reads. The y axis shows the -log10(p-value) of the associations between MT read percent and the technical

factors estimated from the RNA-seq data. The x axis shows the expected -log10(p-value) if no true associations between the MT reads percent and technical factors exist. (B). The intergenic read percent is dominated by the MT read percent, including the MT reads. (C). The qq-plot shows the correlations between the MT read percent and estimated RNA metrics, excluding the MT reads. Compared to A, the associations between the MT read percent and other technical factors are much weaker. (D). The intergenic read percent is not correlated with the MT read percent, excluding the MT reads.
(TIF)

**S1 Table. Characteristics of SN-RNA-seq samples.**
(XLSX)

**S2 Table. Proportions of the 8 cell types identified from 6 sn-RNA-seq adipose samples.**
(XLSX)

**S3 Table. The MuSiC estimated cell-type proportions in the METSIM cohort.**
(XLSX)

**S4 Table. The estimated adipose cell-type proportions are associated with the Matsuda index.**
(XLSX)

**S5 Table. A multi-linear model shows the significant associations between the Matsuda index and the other traits.**
(XLSX)

**S6 Table. The prediction model of Matsuda index.**
(XLSX)

**S7 Table. Phenotypes of the human adipose RNA-seq cohorts.**
(XLSX)

**S8 Table. Summary statistics of the IVs in the MR analysis between obesity (body fat percent) and prediabetes.**
(XLSX)

**S9 Table. Summary statistics of the IVs in the MR analysis between obesity (body fat percent) and Matsuda index.**
(XLSX)

**S10 Table. Signature genes identified in the 8 adipose cell-types.**
(XLSX)

**S11 Table. Technical factors observed in the METSIM adipose RNA-seq data.**
(XLSX)

## Acknowledgments

We thank the individuals who participated in the METSIM, GTEx, FTC, and UK Biobank as well as Jaakko Kaprio and Aila Rissanen for the contributions to the FTC study. We also thank the UNGC sequencing core at UCLA for performing RNA sequencing. The GTEx data used for the analyses described in this manuscript were obtained from the GTEx Portal on 08/14/2016 and dbGAP (accession number phs000424.v6.p1) on 08/11/2016. This research has been conducted using the UK Biobank Resource under application number 33934.

## Author Contributions

**Conceptualization:** Zong Miao, Päivi Pajukanta.

**Data curation:** Marcus Alvarez, Arthur Ko, Yash Bhagat, Sini Heinonen, Linda Liliana Muñoz-Hernandez, Miguel Herrera-Hernandez, Carlos Aguilar-Salinas, Teresa Tusie-Luna, Karen L. Mohlke, Markku Laakso, Kirsi H. Pietiläinen, Päivi Pajukanta.

**Formal analysis:** Zong Miao.

**Funding acquisition:** Päivi Pajukanta.

**Investigation:** Zong Miao, Päivi Pajukanta.

**Methodology:** Zong Miao, Marcus Alvarez, Elior Rahmani, Brandon Jew, Eran Halperin, Päivi Pajukanta.

**Project administration:** Päivi Pajukanta.

**Resources:** Päivi Pajukanta.

**Software:** Zong Miao, Marcus Alvarez.

**Supervision:** Päivi Pajukanta.

**Validation:** Zong Miao, Päivi Pajukanta.

**Visualization:** Zong Miao.

**Writing – original draft:** Zong Miao, Päivi Pajukanta.

**Writing – review & editing:** Zong Miao, Marcus Alvarez, Arthur Ko, Yash Bhagat, Elior Rahmani, Brandon Jew, Sini Heinonen, Linda Liliana Muñoz-Hernandez, Miguel Herrera-Hernandez, Carlos Aguilar-Salinas, Teresa Tusie-Luna, Karen L. Mohlke, Markku Laakso, Kirsi H. Pietiläinen, Eran Halperin, Päivi Pajukanta.

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
