## [Decision Letter · Decision Letter 0]

4 Mar 2020

Dear Dr Pajukanta,

Thank you very much for submitting your Research Article entitled 'The causal effect of obesity on prediabetes and insulin resistance reveals the important role of adipose tissue in insulin resistance' to PLOS Genetics. Your manuscript was fully evaluated at the editorial level and by independent peer reviewers. The reviewers appreciated the attention to an important problem, but raised some substantial concerns about the current manuscript. Based on the reviews, we will not be able to accept this version of the manuscript, but we would be willing to review again a much-revised version. We cannot, of course, promise publication at that time.

If you decide to revise the manuscript for further consideration at PLOS Genetics, please aim to resubmit within the next 60 days, unless it will take extra time to address the concerns of the reviewers, in which case we would appreciate an expected resubmission date by email to plosgenetics@plos.org.

[LINK]

We are sorry that we cannot be more positive about your manuscript at this stage. Please do not hesitate to contact us if you have any concerns or questions.

Yours sincerely,

Elizabeth R. Hauser

Guest Editor

PLOS Genetics

Gregory Barsh

Editor-in-Chief

PLOS Genetics

The reviewers identified three aspects of the manuscript which require major edits and possibly re-analysis. The first is that underlying assumptions about the relationships between BMI, subcutaneous fat, obesity, pre-diabetes and diabetes suggest relationships that might not hold in these datasets. Please describe and defend the rationale for these relationships. Both reviewers questioned the use of a single MR method and suggested a robustness analysis of the assumptions, particularly the assumption of homogeneity and lack of pleiotropy. Finally, the selection of the instrument should be defended and an independence of the selection of the SNPs to avoid bias should be addressed.

Reviewer's Responses to Questions

**Comments to the Authors:**

Reviewer #1: The review is uploaded as an attachment.

Reviewer #2: The aim of the study is to clarify the association between obesity and prediabetes using in part genetic epidemiological and in part cellular approaches. I was not fully comfortable with the way the study was conceptualised. There are also some issues with the methodology used for instrument selection and for the MR analyses in general. Comments below provide further details relating to my concerns.

Author summary was not conceptually clear, and it should be revised. It also illustrates some of the issues with the paper in general. Examples:

“Previous studies have shown that obesity has a causal effect on T2D;however, it remains unknown whether obesity causes prediabetes and insulin resistance already in non-diabetic individuals.”

I have a problem with logic underlying this conceptualisation, as if we know that obesity causes T2D, then obesity has to cause prediabetes in non-diabetic individuals before they progress to T2D. This is not to say that there could not be mechanistic insights we are still to gain in this context, or that elevated blood sugar would could not contribute in part to obesity, but overall I felt very underwhelmed by this conceptualisation.

“Next, we investigated the role of subcutaneous adipose tissue in these obesogenic effects. We discovered that the adipose mitochondrial gene expression and body mass index (BMI) are independently associated with insulin resistance after adjusting for the tissue heterogeneity.”

Here we have a problem as the text implies an interest to look at the role of subcutaneous adipose tissue, and the way they are doing that is in part looking into the associations with BMI. BMI is in no way a reliable indicator for subcutaneous adipose tissue. BMI would incorporate subcutaneous and all other types of adipose tissue, but it also reflects the amount of muscle mass. Indeed, it is known to have a strong gender bias, and muscular men are likely to be classified as ‘obese’ using BMI cut-offs (and even if their subcutaneous fat would be likely to be very limited). UK Biobank does have information on fat-percentage and other more refined indicators of adiposity, which could prodice helpful information in this context.

Related to the above and the overall conceptualisation in the paper. The authors use BMI interchangeably with ‘obesity’ but these are two very different issues. This is particularly concerning in the context of conclusions, where the they make mechanistic claims about the role of adipose tissue in obesity. Overall this study comes across as slightly odd mix of conducting analyses and conceptualisation in a very rough way (BMI=subcutaneous tissue=obesity) and yet aiming to get to more refined analyses and understanding through the cellular approaches.

Methods:

MR PRESSO does not control for pleiotropy, but it includes a test which can help to detect pleiotropy. In general, the advice is to include multiple different MR methods, including pleiotropy robust methods such as MR Eggr and others.

Instrument selection was based on the same dataset in which the effects were tested, which can introduce substantial bias into MR analysis (Burgess et al. 2016, Epidemiology). They investigators further filtered the BMI SNPs based on the observed associations in the METSIM cohort, potentially further aggravating the bias. I suggest the analyses to be revised such that the selection of instruments is based on an independent dataset, such as a signals from genomewide association meta-analyses on adiposity traits, assuming these do not include either of the cohorts used.

They claim that they confirmed that the instruments were not associated with any confounders, but there were no analyses to support this claim. It appears that the authors take this claim based on MR presso, which suggests some confusion about related methods.

**Have all data underlying the figures and results presented in the manuscript been provided?**

Reviewer #1: No: See details in the attachment.

Reviewer #2: None

PLOS authors have the option to publish the peer review history of their article (what does this mean?). If published, this will include your full peer review and any attached files.

Reviewer #1: No

Reviewer #2: No

---

## [Decision Letter · Decision Letter 1]

12 Jul 2020

Dear Dr Miao,

Thank you very much for submitting your Research Article entitled 'The causal effect of obesity on prediabetes and insulin resistance reveals the important role of adipose tissue in insulin resistance' to PLOS Genetics. Your manuscript was fully evaluated at the editorial level and by independent peer reviewers. The reviewers appreciated the attention to an important topic but identified some aspects of the manuscript that should be improved.

We therefore ask you to modify the manuscript according to the review recommendations before we can consider your manuscript for acceptance. Your revisions should address the specific points made by each reviewer.

[LINK]

Yours sincerely,

Elizabeth R. Hauser

Guest Editor

PLOS Genetics

Gregory Barsh

Editor-in-Chief

PLOS Genetics

The two reviewers agreed that the manuscript has been improved. However, each reviewer had a few more comments, mostly regarding clarification and text editing that remain to improve the manuscript. Please consider these suggested edits. One major comment was the role of covariates in interpretation of the MR results and the potential for association with the IV. Please address this comment.

Reviewer's Responses to Questions

**Comments to the Authors:**

Reviewer #1: The review is uploaded as an attachment.

Reviewer #2: The authors have made notable improvements to the analyses. Some further queries remain.

Please test the associations of the IV with confounders, e.g. sex, education, social class, physical activity, smoking, and alcohol. At the moment the paper infers lack of confounder associations from comparisons across MR approaches which is not appropriate.

I would suggest a tidy up of terminology, and an aim to revise to simply refer to things for what they are. If a study has shown something is related to higher BMI, then why not simply refer to association between higher BMI and a given outcome, rather than labelling it ‘obesity’ (represented by BMI)? Obesity is when BMI is >30 (WHO classification), while in a continuous scale a higher BMI can occur across the whole range of BMIs.

If outliers are not detected, MR – presso produces estimates that are identical to IVW, which assumes no horizontal pleiotropy. The authors write that they used MR presso to “correct for pleiotropy and test for causal effects”. How was pleiotropy corrected? If there indeed was no outliers identified what did they correct for? I suggest removing statements relating to “correcting for pleiotropy”. It is good also to appreciate, that none of the MR methods are “perfect” and the absence of evidence for pleiotropy does not necessarily prove the absence of pleiotropy.

**Have all data underlying the figures and results presented in the manuscript been provided?**

Reviewer #1: Yes

Reviewer #2: None

PLOS authors have the option to publish the peer review history of their article (what does this mean?). If published, this will include your full peer review and any attached files.

Reviewer #1: No

Reviewer #2: No

---

## [Editor Report · Decision Letter 2]

29 Jul 2020

Dear Dr Pajukanta,

We are pleased to inform you that your manuscript entitled "The causal effect of obesity on prediabetes and insulin resistance reveals the important role of adipose tissue in insulin resistance" has been editorially accepted for publication in PLOS Genetics. Congratulations!

Yours sincerely,

Elizabeth R. Hauser

Guest Editor

PLOS Genetics

Gregory Barsh

Editor-in-Chief

PLOS Genetics

Comments from the reviewers (if applicable):

**Data Deposition**

http://datadryad.org/submit?journalID=pgenetics&manu=PGENETICS-D-19-01853R2

**Press Queries**

---

## [Editor Report · Acceptance letter]

10 Sep 2020

PGENETICS-D-19-01853R2 

The causal effect of obesity on prediabetes and insulin resistance reveals the important role of adipose tissue in insulin resistance 

Dear Dr Pajukanta, 

We are pleased to inform you that your manuscript entitled "The causal effect of obesity on prediabetes and insulin resistance reveals the important role of adipose tissue in insulin resistance" has been formally accepted for publication in PLOS Genetics! Your manuscript is now with our production department and you will be notified of the publication date in due course.

With kind regards,

Jason Norris

PLOS Genetics

On behalf of:
